# Diastereoselective Synthesis of the Borylated d-Galactose Monosaccharide 3-Boronic-3-Deoxy-d-Galactose and Biological Evaluation in Glycosidase Inhibition and in Cancer for Boron Neutron Capture Therapy (BNCT)

**DOI:** 10.3390/molecules28114321

**Published:** 2023-05-24

**Authors:** Michela I. Simone

**Affiliations:** 1Discipline of Chemistry, University of Newcastle, Callaghan, NSW 2308, Australia; michela_simone@yahoo.co.uk or michela.simone@csiro.au; 2Newcastle CSIRO Energy Centre, 10 Murray Dwyer Circuit, Newcastle, NSW 2304, Australia

**Keywords:** d-galactose, boronic acid, boronate, organic boron, BNCT, glycosidase, Fsp^3^ index, NMR, mutarotation, borarotation

## Abstract

Drug leads with a high Fsp^3^ index are more likely to possess desirable properties for progression in the drug development pipeline. This paper describes the development of an efficient two-step protocol to completely diastereoselectively access a diethanolamine (DEA) boronate ester derivative of monosaccharide d-galactose from the starting material 1,2:5,6-di-*O*-isopropylidene-α-d-glucofuranose. This intermediate, in turn, is used to access 3-boronic-3deoxy-d-galactose for boron neutron capture therapy (BNCT) applications. The hydroboration/borane trapping protocol was robustly optimized with BH_3_.THF in 1,4-dioxane, followed by in-situ conversion of the inorganic borane intermediate to the organic boron product by the addition of DEA. This second step occurs instantaneously, with the immediate formation of a white precipitate. This protocol allows expedited and greener access to a new class of BNCT agents with an Fsp^3^ index = 1 and a desirable toxicity profile. Furthermore, presented is the first detailed NMR analysis of the borylated free monosaccharide target compound during the processes of mutarotation and borarotation.

## 1. Introduction

Glycosidase inhibition and BNCT are disease management strategies we pursue through the design and synthesis of novel chemical entities imbued with capabilities to interact with carbohydrate-active enzymes and through the covalent linkage of organic boron to biologically active substrates [1,2,3,4,5,6,7,8,9].

The chemical behavior of organic boron, principally in its boronic acid (R-B(OH)_2_) and boronate forms (R-B(OR’)(OH), R-B(OR’)_2_, R-B^−^(OH)_4_ and R-B^−^(OR’)_3_), is kaleidoscopic and synthetically challenging, but provides the opportunity to produce glycosidase modulators possessing an expanded interaction profile with enzymes (extending to reversible covalent bonds via the B atom empty *p*-orbital) and switch on/switch off BNCT agents [1,2,3,4,5,6,7]. These capabilities—coupled with the low toxicity of organic boron moieties—make drug leads of this type highly desirable and of paramount importance in medicinal chemistry [1,4]. The development of synthetic methodologies for functionalization of high Fsp^3^ index [1,2,3,4,5,6,7,8,9,10,11,12,13] heteroatom intermediates is crucial to expanding the boundaries of accessible chemical space [14].

This paper communicates the efficient and greener introduction and purification of a boronate ester onto 3-deoxy-d-gluco alkene intermediate **3** (Figure 1). The diastereoselective production of a d-galacto derivative borylated at position C-3 via trapping of the borane intermediate with DEA was achieved in under two minutes in dry conditions. C-3 borylated d-galactose **7** was obtained in four synthetic steps from the starting material diacetone-d-glucose **1** in an overall yield of 53%. This represents the most expedited, robust, greenest, and highest yielding protocol for the installation of boronic acids on high Fsp^3^-index substrates, such as monosaccharides and their derivatives.

## 2. Results

### 2.1. Synthesis

This investigation went hand-in-hand with a parallel study where we developed a synthetic methodology for borylated iminosugars of d*-allo* configuration from the starting material 1,2:5,6-di-*O*-isopropylidene-α-d-allofuranose (which will be discussed in a forthcoming publication). The synthesis described here started with the C-3 epimeric compound diacetone-d-glucose 1 (Figure 1). Triflation of the sole unprotected hydroxyl on C-3 in DCM at 0 °C for 30 min gave unstable intermediate 2. Triflate 2 provided an opportunity to either substitute C-3 or eliminate it to produce an alkene. The alkene could be installed between C-3 and either C-2 or C-4. Elimination was elicited using DBU in refluxing diethyl ether to produce alkene 3, possessing the double bond between C-3 and C-4. Alkene 3 is thermodynamically favored [15,16].

Alkene 3 was produced in 92% yield (over two steps) on a multigram scale via treating triflate 2 with DBU (1.15 eq.). In the literature, the same alkene was produced in 50% yield via treatment of triflate 2 with MeLi [16], in 98% yield with DBU, in 67% yield with potassium *tert*-butoxide [17], and in 84% yield on a multigram scale [18]. Our m.p. and [α]_D_ values are comparable to the literature values (see Experimental section). An extra portion of DBU was required to drive the reaction to completion. Purification of the alkene was achieved via flash column chromatography to afford 3 as a white solid (Appendix A). Crystallization attempts with DCM, hexane, and petroleum ether did not yield a crystalline solid. It was confirmed through 2D NMR experiments that the E2 reaction was regioselective and that the double bond was installed between C-3 and C-4. Two stereocenters were removed in this step. This two-step sequence was efficiently optimized on a multigram scale (91.6% yield).

Similarly highlighted: the numbering system on starting material 1; the two main resonance contributors of alkene 3 and (in the indigo box) the two most likely products 4 of the hydroboration reaction (iii), possessing, respectively, d-*galacto* and d-*allo* absolute stereochemical configurations.

#### 2.1.1. Hydroboration/Borane Trapping

Alkene hydroboration [19] is one such important reaction, permitting the introduction of up to two chiral centers across a double bond in one synthetic step. The hydroboration reaction of the alkene involves the *cis*-delivery of hydride (H:^−^) and BH_2_ across the double bond. This delivery could potentially lead to four different products: the Markovnikov and anti-Markovnikov products, with approaches from either the top- or bottom-face of the THF-ring. The anti-Markovnikov regio isomers, where the hydride and BH_2_ groups are delivered, respectively, on C-4 and C-3 atoms of alkene 3, are predicted to be favored also due to the increased nucleophilicity of C-3, which is β to the THF oxygen (Figure 1, resonant forms of 3). The diastereoselectivity of this addition reaction can either afford the d-*galacto* C-3 borane intermediate 4 (if the new C-B and C-H bonds project out towards the reader) or d-*allo* (if the new C-B and C-H bonds project away from the reader) (Figure 1, indigo box). Regio- and diastereo-isomeric purity can be challenging to achieve. One extra challenge is represented by the development of a purification protocol that would not need silica gel (e.g., flash column chromatography or prep t.l.c.) due to the interactions between silica gel and inorganic and organic B atoms that can lead to product loss [20,21]. Purifications of high Fsp^3^ index boron-bearing derivatives are an ongoing area of research in our research lab, which has spurred developments into green chemistry to reduce or eliminate solvent use and through the removal of purification steps involving silica gel [1,4,22].

The hydroboration/borane trapping two-step reaction sequence and subsequent purification protocol development went through a series of optimizations (Appendix A).

The hydroboration reaction was very sensitive to moisture and had to be carried out in strictly dry conditions. It was ultimately optimized via the investigation of four different temperatures (r.t., 40 °C, 60 °C, 80 °C) with four different numbers of equivalents of BH_3_.THF (1.5, 2.5, 5.0, 10.0) for various time periods (half an hour, 1.5 h, 4.5 h, one day) in THF under an atmosphere of nitrogen and in dry conditions. It was found that the best reaction conditions for borane 4 formation were r.t., 2.5 eq of BH_3_.THF for 1.5 h. Reaction progress was monitored by t.l.c. analysis (acetone/hexane, 1:2), where alkene 3 appeared at R*_f_* 0.60 and borane 4 as one spot streaking across R*_f_* 0.00–0.20. If t.l.c. analysis indicated the reaction had not gone to completion, an additional portion of BH_3_.THF (0.5 mL) was added, and the reaction was allowed to stir for a further 1.5 h. 

Use of other hydroborating agents: 9-BBN [23,24], catecholborane, and BH_3_.SMe_2_ [24,25] did not afford as clean hydroboration reactions, required longer reaction times, resulted in incomplete reactions, and seemingly produced multiple products.

It was ascertained that the hydroboration occurred with complete stereo- and regio-selectivity to give one product 4, for which there is a strong indication that it possesses the d-*galacto* configuration from the *J* coupling values in target compound 7. These have magnitudes aligned with d-*galacto* configuration Karplus angles. For example, the coupling between H-2 and H-3 has a magnitude of 11.4/11.6 Hz, indicating the relative *trans*-arrangement of the two C-H bonds. In 4, it is known that H-2 is pointing away from the acetonide-protecting group. Then it is hypothesized that H-3 is on the same side of the acetonide group, and the newly formed C-B bond is on the same side as H-2. Since hydroboration delivers the H and B atoms in *cis* fashion across a double bond, H-4 should also be pointing in the same direction as H-2. 

Trapping with DEA: Changing the Lewis base from pinacol to DEA provided a successful outcome. The advantage of utilizing DEA lies in the increased polarity of zwitterionic boronate 6, containing tetrahedral B and N atoms with a formal negative charge on the boron and a formal positive charge on the nitrogen. It was reasoned that this educt could also have been more easily re-/crystallized [26]. Under carefully dry conditions, hydroboration of alkene 3 with BH_3_.THF (1 M) formed the corresponding reactive borane intermediate 4 in 1,4-dioxane. This reaction can be efficiently monitored by t.l.c. analysis due to the significant difference between the R*_f_* values of 3 and 4. Borane 4 was then trapped in situ with DEA (3.5 eq.) to form the corresponding organic boron educt 6. Synthetically, as soon as the DEA was added to the stirring solution containing borane 4, a white cloudiness developed, which indicated an instantaneous formation of boronate 6 from 4 (see Appendix A) in under two minutes. The reaction was left to stir for another 15 min. During this time, the reaction mixture’s appearance did not change any further.

The mixture was filtered and dried to give a white solid, which was triturated with diethyl ether and filtered to give product **6** as a white solid. As there was remaining DEA present, the optimized protocol involved dissolving the solid in DCM (20 mL) and washing with HCl (0.1 M, 0.5 mL) to give the pure desired product (170 mg, 58%) in the DCM layer (Appendix A).

Purification protocol: In our experience, the highest yielding protocols involve crystallizations, recrystallizations, and partitioning between solvents [1,4]. In the case of highly polar borylated species, exchange resins tend to provide successful outcomes in our hands. All other methods, when successful, provided low yields of the boron-containing species. This is described in limited detail in the literature, where, for example, purification by flash column chromatography using silica gel (or basic alumina) does not tend to work because the B atoms—as Lewis acids—tend to ‘stick’ to the nucleophilic atoms of silica gel [20,21]. We also found that polar boronic acids interact strongly with silica gel—as in the case of borylated monosaccharides—and can hardly be eluted, even with eluents containing methanol or acetic acid. In some cases, also in our experience, filtration through a short plug of silica can provide the product in moderate yields [27,28], but reproducibility is hard to achieve. For polar species, ion-exchange chromatographic columns work well in our hands and in the literature [29].

On the other hand, the chemical literature shows that highly lipophilic boronic acid derivatives can fairly reliably be purified by flash chromatography on silica [30]. Purification via reverse-phase HPLC [31,32,33] shows partially successful separations, with more hydrophobic (longer C-chain) stationary phases performing better than more hydrophilic ones [20,32,34]. Probably, this is due to the nucleophilic atoms of the stationary phase being more difficult to interact with when contained in more hydrophobic stationary phases. Other HPLC methods have also been developed for the resolution of boron-containing isomers [21,35,36] and tetrahedral boron species [37].

The principal purification attempts are outlined in Appendix A (Appendix A). The aim was to remove the excess DEA along with potential impurities, as at times an additional species was visible in the ^1^H-NMR spectrum. 

In the most successful purification (Purification 3), the reaction mixture (200 mg scale) was filtered to remove the bulk of the excess DEA. This was followed by a trituration with diethyl ether to remove the unidentified compounds, and the solid was dissolved in DCM and partitioned with HCl (0.1 M) to remove the remaining DEA. This method resulted in pure product **6** in the filtrate with a 58% yield.

#### 2.1.2. Deprotection Step

The deprotection step (i.e., removal of the two acetonide groups and DEA) could happen in a step-by-step fashion or globally. A step-by-step protocol would facilitate analysis of the intermediate species and monitoring for structural integrity (Figure 2). 

Three strategies towards the final deprotected product **7** were investigated. Synthetic Strategy 1 was employed in order to ascertain if the unidentified compounds produced in the earlier synthetic step were the same molecule in equilibrium between different species (Figure 2) or different chemical entities. The deprotection of the DEA was carried out in the hope that the ^1^H-NMR spectrum would simplify to one species. This deprotection was achieved by dissolving the product in diethyl ether with the addition of HCl (0.1 M). The reaction was allowed to stir at room temperature for 1 h [38]. The ^1^H-NMR of the reaction mixture indicated the reaction had occurred and appeared to contain fewer species than the initial fraction of intermediate **6** (from the ether filtrate), but the free DEA resulted in an unclear spectrum along with a small amount of potentially multiple species (possibly again from an equilibrium occurring). 

In Synthetic Strategy 2, from pure product **6**, the acetonide on C-5 and C-6 was selectively deprotected. The pure starting material was dissolved in MeOH/H_2_O/AcOH (5:6:4), and the reaction was heated to 50 °C and allowed to stir. A ^1^H-NMR after three hours indicated a reaction was occurring but had not reached completion. The reaction was stirred at 50 °C overnight. A ^1^H-NMR after 21 h still showed the presence of starting material **6**, so the reaction was heated to 70 °C and stirred for an additional 4.5 h. The ^1^H-NMR at this point seemed to indicate no remaining starting material. Purification was attempted by precipitating product **6b** by dissolving the crude residue in methanol with the dropwise addition of acetone, then hexane. A white precipitate crashed out. The solution was filtered, and the solid was redissolved in methanol and concentrated in vacuo (23 mg). This reaction resulted in a mixture of products, as observed by the ^1^H-NMR analysis, which were unable to be separated or characterized. It is hypothesized that the deprotection proceeded further than selective deprotection of the C-5 and C-6 acetonide, and the DEA may have also been deprotected, resulting in a mixture of products at different stages of deprotection.

Synthetic Strategy 3 involved global deprotection using trifluoroacetic acid (TFA). Intermediate **6** (57 mg) was stirred in a solution of TFA/DCM/H_2_O (1:1:0.2) at room temperature for 45 min. Acetonitrile (6 mL) was then added, and a precipitate was formed immediately. The solution was filtered. An extra portion of acetonitrile was added to the reaction solution until no further solids crashed out, followed by filtration. Upon drying in vacuo, the filtrate fraction turned green as a sticky, tough, and amorphous solid (minor fraction). Whereas, upon drying under a stream of nitrogen, the solid fraction went brown (major fraction). Each fraction appeared to contain product **7** by ^1^H-NMR, possibly as a mixture of anomers and boronic acid/intramolecularly derived boronate ester. Although more than one species appears in the NMR spectra, these look remarkably clean, in that four species can be identified as predominant in supposed equilibrium with one another. It is possible that at the time of the NMR data collection, the solution had not reached the mutarotation equilibrium yet.

Free DEA was identified in both fractions. Attempts at crystallizing either **7** or the DEA were unsuccessful during this investigation. Both the solid and filtrate fractions were dissolved in water, and acetone was added. They were allowed to sit at −25 °C overnight, but crystallization did not occur for the filtrate fraction. However, a precipitate formed from the solid fraction. The solution was filtered, and both the solid and filtrate were collected. The filtrate was determined to contain the desired product **7** in a quantitative yield with some minor impurities and free DEA. Further crystallization attempts were unsuccessful.

## 3. Biological Evaluation in Glycosidase Inhibition and in Cancer for BNCT

### 3.1. Glycosidase Assay

Glycosidase modulation studies [4,8,9,39] need to explore selectivity as well as potency. Our drugs and controls are therefore screened against a panel of glycosidases in methanol (Table 1) to identify the glycosidase-related disease area(s), selectivity profile, and potency of biological action for each drug. The presence of organic boron covalently bonded to the drug structure also grants a reduced toxicity profile [1]. Following the glycosidase inhibition range recommendations [39], an IC_50_ value >250 μm denotes weak inhibition, 100–249 μm denotes moderate inhibition, 10–99 μm good inhibition, 0.1–9 μm potent inhibition, and <0.1 μm very potent inhibition. Compounds **6** and **7** are both soluble in water. Our boronated d-galactose **7** represents a novel lead in glycosidase inhibition.

#### 3.1.1. Glycosidases

The glycosidases screened are the following: rice α-glucosidase, yeast α-glucosidase, *Bacillus* α-glucosidase, rat intestinal maltase α-glucosidase, almond β-glucosidase, bovine liver β-glucosidase, coffee beans α-galactosidase, bovine liver β-galactosidase, Jack bean α-mannosidase, snail β-mannosidase, *Penicillium decumbens* α-L-rhamnosidase, bovine kidney α-L-fucosidase, *Eschierichia coli* β-glucuronidase, bovine liver β-glucuronidase, porcine kidney trehalase, *Aspergillus niger* amyloglucosidase, and bovine kidney *N*-acetyl-β-glucosaminidase. The disease areas for each glycosidase are summarized in [1]. 

The biological activities in glycosidase modulation for control compounds and our intermediate **6** and target compound **7** follow in Table 1. 

#### 3.1.2. Controls 

Borocaptate sodium (BSH) and 4-borono-L-phenylalanine (BPA), and their ^10^B-enriched congeners ^10^B-BSH and ^10^B-BPA, are the controls. To our knowledge, these drugs have never been reported in a glycosidase assay, apart from in [1]. BSH and BPA are the drugs currently clinically used in BNCT. It is possible to see that none of them significantly inhibit any of the glycosidases in the panel at 100 or 1000 μM. In the panel in Table 2, percent inhibitions range from a minimum value of 0 to a maximum value of 19.6. 

#### 3.1.3. Intermediate 6 and Target Compound **7**

Intermediate **6** and target compound **7** drugs presented in this work do not provide any appreciable degree of inhibition. However, it is interesting to notice that bovine liver β-galactosidase is the glycosidase inhibited the most, with a percent inhibition value of 40.7. At this stage, it is unclear which anomer/s and conformation/s would be present in the active site of each enzyme. REF However, preferred inhibition for the β-galactosidase may point to a (β-)galactose conformation of target compound **7**. Bovine liver β-glucuronidase also displays a degree of inhibition (32.4%), possibly due to compound **7** sitting in such a way as to display the boronic acid in the area where the carboxylic acid of glucuronic acid would sit. A 33.8% inhibition against rice α-glucosidase is also noticed. 

### 3.2. Cancer Assay and Structure Activity Relationships 

In our laboratory, we develop novel cancer radiotherapy agents for BNCT, which we see as a broad-spectrum approach to cancer management. It would be advantageous prior to irradiation if the boron-containing drugs accumulated more selectively in cancer cells vs. healthy cells [40,41]. If the drugs do not accumulate selectively, the delivery of radiation is required with greater accuracy. 

BNCT is a non-invasive and least destructive radiation therapy currently available [42,43,44,45,46,47]. The use of a borylated drug in BNCT would ideally require that it be non-toxic in the absence of radiation. Following two studies of the development of novel families of BNCT candidates via the development of synthetic protocols, purification protocols, and toxicity studies, we report here one further drug lead as a potential BNCT agent. Target compound **7**—being a monosaccharide analogue—has the potential to exploit the same entry mechanisms into cells (healthy and cancer) as natural monosaccharides, much like 2-fluodeoxyglucose does [48,49]. 2-Fluodeoxyglucose has the capability to exploit the Warburg effect (this implies cells find it difficult to discriminate this synthetic structure from natural monosaccharide structures) for selective entry into cancer versus healthy cells [49,50,51], to accumulate in cells after phosphorylation at C-6 (again successfully exploiting enzymes that chemically modify natural monosaccharide substrates) [52,53], and to cross the blood-brain barrier [54]. However, a number of challenges still need to be overcome [43,44,45,55]. Target compound **7** could also allow for the accumulation of the borylated monosaccharide into cancer cells, priming them for irradiation with slow neutrons during BNCT and allowing for the management of cancers that are difficult to reach and treat, such as brain tumors. Similarly, in this area, candidate BNCT agents with lower intrinsic toxicity, a more favorable pharmacokinetic profile, and a high Fsp^3^ index should be more likely to reach the clinic. 

We focus on synthetic developments for the introduction of organic boron into substrates to produce novel BNCT candidates. It has long been known that organic boron is an essential element for plants [56,57] and is likely to be essential for human and animal health [58]. A comparison of toxicological data for organic boron-containing compounds with their non-borylated parent compounds shows that the presence of organic boron lowers toxicity profiles. For example, benzene LD_50_= 125 mg/kg (human, oral) [59] and an LCLO = 20,000 ppm (human, 5 min); it is carcinogenic and possibly mutagenic [60]. The NIOSH permissible exposure limit for benzene is 1 ppm, the recommended exposure limit is 0.1 ppm, and the immediately dangerous to life and health concentration is 500 ppm [61]. On the other hand, phenylboronic acid has an LD_50_= 740 mg/kg (rat, oral) [62] with no entry for RTECS, ACGIH, IARC, or NTP.

If a BNCT agent also has growth inhibition capability against cancer cells [1], then it is important to screen them in more complex biological systems, such as spheroids, as we conducted recently [40,41].

Intermediate **6** and target compound **7** do not inhibit the growth of the cancer cell lines (Table 2) screened. However, they also do not inhibit the growth of the normal cell line. Their GI_50_ in all cell lines is greater than 50 µM. This is a very promising set of data because it indicates that borylated monosaccharides of the type described herein have the propensity to possess low toxicities. If—as in previous work—these analogues also display a selectivity for cancer cells via exploitation of the Warburg effect [40,41], as FDG does [63,64], then they would become toxic only during the geographically targeted irradiation during BNCT and obliterate almost exclusively cancer cells. Therefore, **7** is a very promising BNCT candidate. 

## 4. Materials and Methods

### 4.1. Glycosidase Inhibition Experimental (Table 1) 

The enzymes α-glucosidase (from yeast), β-glucosidases (from almond and bovine liver), α-galactosidase (from coffee beans), β-galactosidase (from bovine liver), α-mannosidase (from Jack bean), β-mannosidase (from snail), α-L-rhamnosidase (from *Penicillium decumbens*), α-L-fucosidase (from bovine kidney), trehalase (from porcine kidney), β-glucuronidases (from *E. coli* and bovine liver), amyloglucosidase (from *A. niger*), *para*-nitrophenyl glycosides, and various disaccharides were purchased from Sigma-Aldrich Co. 

Brush border membranes were prepared from the rat small intestine according to the method of Kessler et al. [65] and were assayed at pH 6.8 for rat intestinal maltase using maltose. For rat intestinal maltase, porcine kidney trehalase, and *A. niger* amyloglucosidase activities, the reaction mixture contained 25 mM maltose and the appropriate amount of enzyme, and the incubations were performed for 10–30 min at 37 °C. The reaction was stopped by heating at 100 °C for 3 min. After centrifugation (600 g; 10 min), the resulting reaction mixture was added to the Glucose CII-test Wako (Wako Pure Chemical Ind., Osaka, Japan). The absorbance at 505 nm was measured to determine the amount of the released d-glucose. Other glycosidase activities were determined using an appropriate *para*-nitrophenyl glycoside as substrate at the optimum pH of each enzyme. The reaction mixture contained 2 mM of the substrate and the appropriate amount of enzyme. The reaction was stopped by the addition of 400 mM Na_2_CO_3_. The released *para*-nitrophenol was measured spectrometrically at 400 nm. All reactions run in methanol.

### 4.2. Cancer Screening Experimental (Table 2)

All test agents were prepared as stock solutions (20 mM) in dimethyl sulfoxide (DMSO) and stored at −20 °C. Cell lines used in the study included: HT29 (colorectal carcinoma), U87 and SJ-G2 (glioblastoma), MCF-7 (breast carcinoma), A2780 (ovarian carcinoma), H460 (lung carcinoma), A431 (skin carcinoma), Du145 (prostate carcinoma), BE2-C (neuroblastoma), MiaPaCa-2 (pancreatic carcinoma), SMA560 (spontaneous murine astrocytoma), and ADDP (Cis Res Ovarian), together with one non-tumor-derived normal breast cell line (MCF10A). All cell lines were incubated in a humidified atmosphere with 5 % CO_2_ at 37 °C. The cancer cell lines were maintained in Dulbecco’s modified Eagle’s medium (DMEM; Sigma, Australia) supplemented with fetal bovine serum (10%), sodium pyruvate (10 mM), penicillin (100 IUmL^−1^), streptomycin (100 µg mL^−1^), and L-glutamine (2 mM).

The non-cancer MCF10A cell line was maintained in DMEM:F12 (1:1) cell culture media, 5% heat inactivated horse serum, and supplemented with penicillin (50 IUmL^−1^), streptomycin (50 µg mL^−1^), HEPES (20 mM), L-glutamine (2 mM), epidermal growth factor (20 ng mL^−1^), hydrocortisone (500 ng mL^−1^), cholera toxin (100 ng mL^−1^), and insulin (10 mg mL^−1^).

Growth inhibition was determined by plating cells in duplicate in medium (100 µL) at a density of 2500–4000 cells per well in 96-well plates. On day 0 (24 h after plating), when the cells are in logarithmic growth, medium (100 µL) with or without the test agent was added to each well. After 72 h of drug exposure, growth inhibitory effects were evaluated using the (3-(4,5-dimethyltiazol-2-yl)-2,5-diphenyltetrazolium bromide) (MTT) assay and absorbance read at 540 nm. The percentage growth inhibition was calculated at a fixed concentration of 25 µM, based on the difference between the optical density values on day 0 and those at the end of drug exposure. Each data point is the mean ± the standard error of the mean (SEM) calculated from three replicates, which were performed on separate occasions and in separate cell line passages. 

### 4.3. Chemistry 

#### General Experimental

Numbering system: Spectroscopic data for all compounds is assigned based on a numbering system derived from systematic naming of materials according to IUPAC recommendations on carbohydrate nomenclature [66]. The numbering is given in Figure 1 by the red numbers on structure **1.**

Solvents: Dichloromethane (DCM), 1,4-dioxane, and pyridine were purchased from the Aldrich Chemical Company in sure-seal^TM^ reagent bottles. Water was distilled. All other solvents (analytical or HPLC grade) were used as supplied without further purification. Deuterated acetone (acetone-d^6^), chloroform (CDCl_3_), methanol (MeOD), and water (D_2_O) were used as NMR solvents.

Reagents: Reactions performed in dry conditions and under an atmosphere of nitrogen were maintained by an inflated balloon. Borane.tetrahydrofuran (BH_3_.THF, 1M) was purchased from Sigma-Aldrich in sure-seal^TM^ bottles. The reagents were used as provided without further purification, with NMR analysis confirming an acceptable degree of purity and correct structural identity.

Purification via silica gel column chromatography was performed on Davisil 40–63 micron silica gel. Thin layer chromatography (t.l.c.) was performed on aluminum sheets coated with 60 F254 silica by Merck and visualized using the UVG-11 Compact UV lamp (254 nm) or stained with a solution of 12.0 g ammonium molybdate and 0.5 g ceric ammonium molybdate in 15 mL of concentrated sulfuric acid and 235 mL of distilled water.

Nuclear magnetic resonance (NMR) spectra were recorded on the Bruker Ascend^TM^ 400 in the deuterated solvent stated. Chemical shifts (δ) are quoted in ppm and coupling constants (*J*) in Hz. Residual signals from CDCl_3_ (7.26 ppm for ^1^H-NMR and 77.16 ppm for ^13^C-NMR), deuterated acetone (2.05 ppm for ^1^H-NMR and 29.84 ppm for ^13^C-NMR), deuterated methanol (3.31 ppm for ^1^H-NMR and 49.00 ppm for ^13^C-NMR), and deuterium oxide (4.89 ppm for ^1^H-NMR) were used as an internal reference [67]. NMR spectra in the Appendix A were produced using TopSpin 4.2.0.

Infrared (IR) spectroscopy: IR spectra were obtained on a PerkinElmer Spectrum Two Spectrometer, on a PerkinElmer Spectrum 2 with UATR (intermediates **3** and **6**), and on a Bruker Invenio R (target compound **7**). Only characteristic peaks are quoted and in units of cm^−1^.

Low resolution mass spectrometry (LRMS) spectra were obtained on an Agilent Technologies 1260 Infinity UPLC system with a 6120 Quadrupole LC/MS in electrospray ionization (ESI) positive and negative modes. All LCMS methods used a mobile phase A of 100% water with 0.1% formic acid and a mobile phase B of 9:1*v/v*ACN/water with 0.1% formic acid. 

High resolution mass spectrometry (HRMS) spectra were obtained from samples suspended in acetonitrile (1 mL with 0.1% formic acid at a concentration of ~1 mg/mL, before being further diluted to ~10 ng/μL in 50% acetonitrile/water containing 0.1% formic acid). Samples were infused directly into the HESI source of a Thermo Scientific Q Exactive™ Plus Hybrid Quadrupole-Orbitrap™ Mass Spectrometer using an on-board syringe pump at 5 μL/min. Data were acquired on the QE+ in both positive and negative ion modes at a target resolution of 70,000 at 200 *m/z*. The predominant ions were manually selected for MS/MS fragmentation (collision energies were altered for each compound to obtain sufficient fragmentation). The data analysis of each sample was performed manually using the Thermo Qual browser, while the isotopic patterns of the predicted chemical formula were modeled using the Bruker Compass Isotope Pattern.

Optical rotations were carried out on a Jasco P-2000 polarimeter with a length of 1.0 dm and a wavelength of 289 nm. Concentrations are quoted in g/mL.

Melting points were taken on a Dynalon SMP100 Digital Melting Point Device and are uncorrected.

### 4.4. Syntheses

#### 4.4.1. 3-Deoxy-1,2;5,6-di-*O*-isopropylidene-*α*-d-erythro-Hex-3-Enofuranose **3**

Trifluoromethanesulfonic anhydride (4.2 mL, 25.01 mmol) was added dropwise to a stirring solution of 1,2:5,6-di-*O*-isopropylidene-α-d-glucofuranose **1** (5.000 g, 19.21 mmol) in dry DCM (25 mL) and pyridine (3.1 mL, 38.49 mmol) at 0 °C under an atmosphere of nitrogen. After 30 min, t.l.c. analysis (EtOAc/hexane, 1:4) showed the formation of one product (R*_f_* 0.60) and complete consumption of the starting material (R*_f_* 0.00–0.10). The reaction was allowed to warm up to room temperature and washed with an aqueous hydrochloric acid solution (1M, 40 mL). The organic layer was concentrated down to give the 1,2:5,6-di-*O*-isopropylidene-3-*O*-trifluoromethanesulfonate-α-d-glucofuranose intermediate **2** (7.088 g) as a yellow crystalline solid, which was immediately reacted without further purification and had the same characterization data as in the literature [16,68]. 1,8-Diazabicyclo(5.4.0)undec-7-ene (4.3 mL, 28.75 mmol) was carefully added to a stirring solution of the triflate intermediate **2** (7.088 g, 18.08 mmol) in diethyl ether (200 mL) at room temperature and refluxed for 19 h. T.l.c. analysis (EtOAc/hexane, 1:4) showed the formation of one product (R*_f_*0.55) and residual starting material (R*_f_*0.60). An extra portion of 1,8-diazabicyclo(5.4.0)undec-7-ene (4.3 mL) was added to the stirring mixture and refluxed for a further 26 h. T.l.c. analysis confirmed complete consumption of the starting material. Partitioning of the reaction mixture was affected with water/diethyl ether (1:10, *v*/*v*) and the aqueous layer was further extracted with diethyl ether (3 × 100 mL). The organic layers were concentrated and purified by flash column chromatography (EtOAC/hexane, 1:10) to afford 3-deoxy-1,2;5,6-di-*O*-isopropylidene-α-d-*erythro*-hex-3-enofuranose **3** (4.264 g, 91.6% over two steps) as a runny pale-yellow oil that solidified upon mechanical stress. M.p. 46–48 °C [Lit m.p. 49–50 °C [16], 48–50 °C [17,69,70], mp 48–49 °C [18], 50–52 °C [71] and 50 °C [72]. *m/z* (LCMS ES^+^): found mass 243.2 [M+H^+^]^+^, required mass 243.1 [M+H^+^]^+^; HRMS (FTMS, ESI): [M+H^+^]^+^ required mass 243.1227 (100%), 244.1261 (13%), 245.1281(2%); Found mass: 243.1228 (100%), 244.1262 (12%), 244.8786 (2%); [α]_d_^23^.^6^ +31.7° (c 0.0018 g/mL, acetone) (Lit [α]_d_^23^ +28° (c 2.7, CHCl_3_) [16], +25.4° (*c* 1.5, CHCl_3_) [18], +21.2° (*c* 1, CHCl_3_) [69,70], 19.8 (c 0.8, ethanol) [70]). ν_max_ (thin film, cm^−1^): 2995, 2978, 2941, 2910, 2885 (m. alkyl C-H), 1670 (m, C=C), 1209 (s, C-O-C). δ_H_ (Acetone-d_6_, 400 MHz): 6.10 (1H, d, *J*_H-1,H-2_ 5.3 Hz, H-1), 5.29 (1H, ddd, *J*_H-2,H-1_ 5.2 Hz, *J*_H-2,H-3_ 2.3 Hz, *J*_H-2,H-5_ 1.4 Hz, H-2), 5.21 (1H, dd, *J*_H-3,H-2_ 2.3 Hz, *J*_H-3,H-5_ 1.1 Hz, H-3), 4.59 (1H, app-tt, *J*_H-5,H-6_ 6.8 Hz, *J*_H-5,H-6′_ 5.7 Hz, *J*_H-5,H-3_ 1.1 Hz, H-5), 4.13 (1H, dd, *J*_H-6,H-6′_ 8.3 Hz, *J*_H-6,H-5_ 6.9 Hz, H-6), 3.89 (1H, dd, *J*_H-6′,H-6_ 8.3 Hz, *J*_H-6′,H-5_ 5.6 Hz, H-6′), 1.40, 1.39, 1.36, 1.32 (12H, 4 × s, 4 × CH_3_); δ_c_ (Acetone-d_6_, 100 MHz): 161.0 (C-4), 112.5, 110.6 (2 × C_quat_ acetonides), 107.6 (C-1), 100.0 (C-3), 84.1 (C-2), 72.1 (C-5), 67.4 (C-6), 28.6, 28.2, 26.6, 25.7 (4 × CH_3_). 

δ_H_ (CDCl_3_, 400 MHz): 6.08 (1H, d, *J*_H-1,H-2_ 5.2 Hz, H-1), 5.30 (1H, dt, *J*_H-2,H-1_ 5.2 Hz, *J*_H-2,H-3_ 1.0 Hz, H-2), 5.25 (1H, dd, *J*_H-3,H-2_ 1.5 Hz, *J*_H-3,H-5_ 0.7 Hz, H-3), 4.59 (1H, app-td, *J*_H-5,H-6′_ 4.5 Hz, *J*_H-5,H-6_ 3.9 Hz, *J*_H-5,H-3_ 0.8 Hz,H-5), 4.15 (1H, dd, *J*_H-6′,H-6_ 5.6 Hz, *J*_H-6′,H-5_ 4.5 Hz, H-6′), 3.97 (1H, dd, *J*_H-6,H-6′_ 5.6 Hz, *J*_H-6,H-5_ 3.8 Hz, H-6), 1.47 (6H, s, 2 × CH_3_), 1.45, 1.39 (6H, 2 × s, 2 × CH_3_). δ_c_ (CDCl_3_, 100 MHz): 160.1 (C-4), 112.3, 110.4 (2 × C_quat_ acetonides), 106.6 (C-1), 99.0 (C-2), 83.4 (C-3), 71.3 (C-5), 67.0 (C-6), 28.3, 27.9, 26.2, 25.5 (4 × CH_3_).

#### 4.4.2. 3-Deoxy-3-Boronodiethanolamine-1,2:5,6-Di-*O*-isopropylidene-α-d-galactofuranose **6**

Under dry conditions and a nitrogen atmosphere, 1,2:5,6-di-*O*-isopropylidene-3-deoxy-α-d-*erythro*-hex-3-enofuranose **3** (202 mg, 0.8255 mmol) was dissolved in 1,4-dioxane (20 mL) and BH_3_.THF (1 M, 2 mL, 2.06 mmol) was added. The reaction was allowed to stir at room temperature for 1.5h, after which TLC indicated, the reaction had not gone to completion, so an additional portion of BH_3_THF (0.5 mL) was added and the reaction stirred for 1.5 h. T.l.c. then showed the reaction had gone to completion with only one spot seen, on the baseline. DEA (297.5 mg, 2.83 mmol) was added, and a white precipitate formed immediately. The reaction was stirred for an additional 15 min. The solution was filtered, and the filtrate was evaporated in vacuo to give a white solid, which was triturated with diethyl ether and filtered to give a white solid. As there was still excess DEA present in the solid, it was dissolved in DCM (20 mL) and washed with 0.1 M HCl (0.5 mL) to give the 3-deoxy-3-boronodiethanolamine-1,2:5,6-di-*O*-isopropylidene-α-d-galactofuranose **6** (169.9 mg, 0.476 mmol, 57.6%) in the DCM layer as a white residue. *m/z* (LCMS ES^+^): found mass 376.2 [M+H^+^+H_2_O]^+^, required mass 376.2 [M+H^+^+H_2_O]^+^; [α]_d_^19^.^1^ —0.14 (c. 0.0148 g/mL, MeOH); ν_max_ (thin film, cm^−1^): 3393 (broad, NH), 3127 (NH), 2987, 2915 (m, alkyl CH), 2745 (m, N-alkyl CH), 1639 (broad weak, NH bend), 1458 (m, B-N), 1372 (m, B-O), 1211, 1159 (s, C-N), 1058 (s, C-B); δ_H_ (CD_3_OD, 400 MHz): 5.71 (1H, d, *J*_H-1,H-2_ 3.5 Hz, H-1), 4.52 (1H, app-d, *J*_H-2,H-1_ 3.4 Hz, H-2), 4.51–4.43 (1H, m, H-5), 4.04 (1H, dd, *J*_H-4,H-5_ 8.4 Hz, *J*_H-4,H-3_ 3.1 Hz, H-4), 3.96 (1H, dd, *J*_H-6′,H-6_ 8.0 Hz, *J*_H-6′,H-5_ 6.6 Hz, H-6′), 3.94–3.88 (1H, m, obscured, H_A’_ DEA), 3.91 (1H, app-dd, *J*_HB’,HA’/D’_ 9.9 Hz, *J*_HB’,HC’_ 5.5 Hz, H_B’_ DEA), 3.83 (1H, ddd, *J*_HA,HB_ 9.6 Hz, *J*_HA,HC_ 7.0 Hz, *J*_HA,HD_ 2.7 Hz, H_A_ DEA), 3.75 (1H, ddd, *J*_HB,HA_ 9.9 Hz, *J*_HB,HC_ 6.5 Hz, *J*_HB,HD_ 4.8 Hz, H_B_ DEA), 3.67 (1H, t, *J*_H-6,H-6′/5_ 7.8 Hz, H-6), 3.28 (1H, ddd, partially obscured, *J*_HC,HD_ 12.0 Hz, *J*_HC,HA_ 7.6 Hz, *J*_HC,HB_ 6.7 Hz, H_C_ DEA), 3.21 (1H, ddd, *J*_HD’,HC’_ 12.0 Hz, *J*_HD’,HB’_ 10.1 Hz, *J*_HD’,HA’_ 6.9 Hz, H_D’_ DEA), 2.94 (1H, ddd, partially obscured, *J*_HD,HC_ 12.1 Hz, *J*_HD,HB_ 5.2 Hz, *J*_HD,HA_ 2.5 Hz, H_D_ DEA), 2.90 (1H, dt, *J*_HC’,HD’_ 12.1 Hz, *J* 5.3 Hz, H_C’_ DEA), 1.52, 1.39, 1.33, 1.29 (4 × 3H, 4 × s, 4 × CH_3_ acetonide), 1.20 (1H, app-d, *J*_H-3,H-4_ 2.9 Hz, H-3); δ_c_ (CD_3_OD, 100 MHz): 112.9, 110.7 (2 × C_quat_ acetonides), 108.2 (C-1), 86.5 (C-4), 85.7 (C-2), 80.3 (C-5), 67.2 (C-6), 63.8 (C_A’B’_ DEA), 63.7 (C_AB_ DEA), 52.7 (C_C’D’_ DEA), 52.5 (C_CD_ DEA), 37.9–35.2 (broad, C-3), 27.8, 27.1, 26.4, 25.7 (4 × CH_3_); δ_B_ (CD_3_OD, 128 MHz): 30.9 (integration: 0.02), 17.2 (integration: 0.03), 10.3 (integration: 1.00).

#### 4.4.3. 3-Boronic-3-Deoxy-d-galactose **7**

3-Deoxy-3-boronodiethanolamine-1,2:5,6-di-*O*-isopropylidene-α-d-galactofuranose **6** (56.5 mg, 0.158 mmol) was stirred at room temperature in a solution of TFA (0.4 mL), DCM (0.4 mL), and H_2_O (0.08 mL) for 45 min. At this time, acetonitrile (6 mL) was added, resulting in the formation of a white precipitate, which was filtered to give two fractions (solid and filtrate). Both fractions were observed to contain product **7** by NMR, although the solid sample appeared purer. Both fractions were dissolved in water and, after the addition of acetone, stored at −25 °C overnight for crystallization. The filtrate showed no precipitate, while a precipitate was formed from the solid, which was filtered. The product was observed in the filtrate. The solvent was evaporated to give 3-boronic-3-deoxy-d-galactose **7** (47.9 mg, 0.230 mmol, quant. with residual free DEA). [α]_D_^19^.^0^ +0.26 (c. 0.0039 g/mL, MeOH); ν_max_ (thin film, cm^−1^): 3315 (broad, OH), 2930 (alkyl CH), 1387 (B-O), 1062 (C-B); Anomeric form composition at time of NMR: α-pyr:β-pyr:α-fur:β-fur ~0.25:1:1:1; * means tentative assignment. α-Fur: δ_H_ (D_2_O, 400 MHz): 5.25 (1H, d, *J*_H-1,H-2_ 4.6 Hz, H-1, α-fur), 4.31 (1H, dd, *J*_H-2,H-3_ 11.3 Hz, *J*_H-2,H-1_ 4.3 Hz, H-2, α-fur), 4.13 (1H, dd, *J*_H-4,H-5_ 5.2 Hz, *J*_H-4,H-3_ 10.8 Hz, H-4, α-fur), 3.75 (1H, obscured by overlapping signals, H-5, α-fur)*, 3.75–3.60 (2H, obscured by overlapping signals, H-6 and H-6′, α-fur), 1.68 (1H, partially obscured app-t, *J*_H-3,H-2_ 11.6 Hz, *J*_H-3,H-4_ 10.8 Hz, H-3, α-fur); δ_c_ (D_2_O, 100 MHz): 94.4 (C-1, α-fur), 80.6 (C-4, α-fur), 74.7 (C-2, α-fur), 73.1 (C-5, α-fur), 62.9 (C-6, α-fur), 32.3–31.3 (C-3, α-fur). β-fur: δ_H_ (D_2_O, 400 MHz): 5.27 (1H, d, *J*_H-1,H-2_ 2.3 Hz, H-1, β-fur), 4.40 (1H, dd, *J*_H-4,H-3_ 8.3 Hz, *J*_H-4,H-5_ 4.1 Hz, H-4, β-fur), 4.27 (1H, dd, *J*_H-2,H-3_ 5.6 Hz, *J*_H-2,H-1_ 2.3 Hz, H-2, β-fur), 3.70 (1H, obscured by overlapping signals, H-5, β-fur)*, 3.75–3.60 (2H, obscured by overlapping signals, H-6 and H-6′, β-fur), 1.68 (1H, partially obscured dd, *J*_H-3,H-4_ 7.9 Hz, *J*_H-3,H-2_ 5.3 Hz, H-3, β-fur); δ_c_ (D_2_O, 100 MHz): 102.2 (C-1, β-fur), 80.2 (C-4, β-fur), 78.7 (C-2, β-fur), 73.9 (C-5, β-fur)*, 36.8–35.0 (C-3, β-fur), 62.7 (C-6, β-fur). α-Pyr: δ_H_ (D_2_O, 400 MHz): 5.24 (1H, partially obscured d, H-1, α-pyr), 4.16–4.10 (1H, obscured by overlapping signals, H-2, α-pyr), 4.14–4.04 (1H, partially obscured, broad s, H-4, α-pyr), 3.85–3.60 (1H, obscured by overlapping signals, H-5, α-pyr)*, 3.69–3.61 (2H, obscured by overlapping signals, H-6 and H-6′, α-pyr), 1.50–1.43 (1H, partially obscured by overlapping signals, H-3, α-pyr); δ_c_ (D_2_O, 100 MHz): 91.2 (C-1, α-pyr), 71.9 (C-5, α-pyr), 71.4–70.4 (C-4, broadened, 104 Hz, α-pyr), 65.0 (C-2, α-pyr), 64.8 (C-6, α-pyr), *not resolved* (C-3, α-pyr). β-pyr: δ_H_ (D_2_O, 400 MHz): 4.62 (1H, d, *J*_H-1,H-2_ 7.8 Hz, H-1, β-pyr), 4.12–4.08 (1H, broad s, partially obscured, H-4, β-pyr ), 3.79 (1H, dd, *J*_H-2,H-3_ 11.4 Hz, *J*_H-2,H-1_ 7.9 Hz, H-2, β-pyr), 1.47 (1H, partially obscured dd, *J*_H-3,H-2_ 11.6 Hz, *J*_H-3,H-4_ 2.6 Hz, H-3, β-pyr), 3.73–3.81 (2H, obscured by overlapping signals, H-6 and H-6′, β-pyr), 3.78 (1H, obscured by overlapping signals, H-5, β-pyr)*; δ_c_ (D_2_O, 100 MHz): 98.6 (C-1, broadened, 15 Hz, β-pyr), 79.9–79.3 (C-5, broadened, 62 Hz, β-pyr), 68.0 (C-2, broadened, 14 Hz, β-pyr), 67.7–67.0 (C-4, broadened, 71 Hz, β-pyr), 32.3–31.3 (C-3, β-pyr), 61.3 (C-6, broadened, 18.5 Hz, β-pyr). 

δ_B_ (D_2_O, 128 MHz): 31.7 (integration: 4.73), 19.4 (integration 1.00). 

## 5. Conclusions

An efficient, expedited, robust, and completely diastereoselective two-step synthesis of a C-3 borylated d-galactose derivative (Fsp^3^ index = 1) was achieved via the development and optimization of a one-pot, two-reaction sequence hydroboration/borane trapping with DEA. This was followed by an expedited and greener purification protocol to give the first borylated d-galactose. This synthetic development opens up the avenue of borylated monosaccharides that can be studied in a cancer context as more effective BNCT agents. These provide the opportunity to achieve more selective uptakes into cancer versus healthy cells, much like FDG does, and to exploit the Warburg effect to gain entry into cancer cells and hexokinase-catalyzed C-6 phosphorylation. This paper also provides a methodology to carefully characterize these new classes of drug leads, principally by NMR, highlighting challenges, providing a body of data for comparison purposes, and a trampoline for further characterization efforts, including computational.

## Data Availability

Data are available from the author.

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
