# Peer review of "Diastereoselective Synthesis of the Borylated d-Galactose Monosaccharide 3-Boronic-3-Deoxy-d-Galactose and Biological Evaluation in Glycosidase Inhibition and in Cancer for Boron Neutron Capture Therapy (BNCT)"

_molecules, 2023, doi:10.3390/molecules28114321_

Round 1

Reviewer 1 Report

The volume of the paper is impressive, although its structure is somewhat hard to comprehend. It is advised to restructure the text, so it is in terms with IMRAD system.

Although it is undoubtably that a lot of thorough investigations have been carried out -  optimization of all synthetic steps, purification protocols, analyzing aim and side products, studying of biological activity of obtained borylated monosaccharide. In this regard, @conclusion@ part seems underwhelming. It is advised to complete this part and give clear statements on what's new in this work - synthetic route, higher yields, better biological activity etc.  

A few questions arise while reading this manuscript.

1. What was the purpose of obtaining 11B NMR spectra for compounds listed in Table 3 if all data is available in the literature already?

2. The reasoning behind glycosidase screening is unclear. If the obtained compound is thought to act as boron-containing agent for BNCT, it is more informative to test cytotoxicity.

There are a few minor writing inaccuracies. 

p9 line 302

triple question marks

p4 line 127

see video in Supplementary Information, but there is no video in SI

p17 line 531

List starts with 3

In conclusion, this is a big work with a lot of scrupulous research done, therefore it is deserved to be published after a minor revision.

Author Response

I would like to thank the Reviewer for doing such an amazingly detailed review work. It’s massively appreciated. I have modified the manuscript according to their suggestions, including dividing the manuscript up into two manuscripts to allow an easier read and focus on Synthesis and Bio in one manuscript and on Spectroscopic Characterisation in another. This division allows for a more focused approach and the reader doesn’t get lost in all the information. Thank you for this

Please see below the answers to the points raised by the Reviewers.

The volume of the paper is impressive, although its structure is somewhat hard to comprehend. It is advised to restructure the text, so it is in terms with IMRAD system. I have added more numbered headings.

Although it is undoubtably that a lot of thorough investigations have been carried out -  optimization of all synthetic steps, purification protocols, analyzing aim and side products, studying of biological activity of obtained borylated monosaccharide. In this regard, @conclusion@ part seems underwhelming. It is advised to complete this part and give clear statements on what's new in this work - synthetic route, higher yields, better biological activity etc.  This has been extended and clarified.

A few questions arise while reading this manuscript.

  1. What was the purpose of obtaining 11B NMR spectra for compounds listed in Table 3 if all data is available in the literature already? Not all of these data are available in the literature. The purpose is to provide a body of relevant B-NMR data for comparison.
  2. The reasoning behind glycosidase screening is unclear. If the obtained compound is thought to act as boron-containing agent for BNCT, it is more informative to test cytotoxicity. These are monosaccharide analogues , so testing for glycosidase modulation is quite crucial, in order to gain insight into potential side-effects for BNCT. However one doesn’t have to necessarily use these borylated monosaccharides for BNCT. If they effectively inhibit glycosidases, they can be glycosidase inhibitors.

There are a few minor writing inaccuracies. 

p9 line 302

triple question marks Removed

p4 line 127

see video in Supplementary Information, but there is no video in SI. I will send it to the editor I am in contact with.

p17 line 531

List starts with 3. I am sorry, I am not sure what this is referring to.

In conclusion, this is a big work with a lot of scrupulous research done, therefore it is deserved to be published after a minor revision.

Reviewer 2 Report

The diethanolamine boronate ester derivative of monosaccharide D-galactose was diastereoselectively synthesized through the efficient one-pot two-step procedures, offering a quicker pathway to obtain the desired borylated product with a greener purification process. The resulting product was characterized in detail and biological studies were conducted to evaluate the inhibitory activity of the synthesized borylated product on glycosidase enzymes. The results showed that the product did not exhibit any significant inhibition of glycosidase enzymes. However, the borylated product demonstrated low cytotoxicity against tumor and normal cell lines and was identified as a promising candidate for cancer treatment using BNCT.

Comments.

1.     The introduction is rather broad, a more focused statement that highlights the importance of glycosidase modulators against diseases would allow readers to better understand the importance of the borylated compound. Additionally, an explanation of how switch on/switch off BNCT agents mediate therapy would enhance the readers' comprehension.

2.     The study claims that the synthetic method in this work achieves the highest yield for the installation of boronic acids on high Fsp3 index substrates. To support this claim, the authors should provide comparative data with conventional methods.

3.     Topic 2.1, I would suggest shortening this section by moving details about purification, storing methods, and solvent choices to the Supporting Information (SI) section. This will help streamline the main text and allow readers to focus on the key aspects of your research.

4.     Topic 2.1.1, trapping optimization, the author provided valuable background information. However, it would be beneficial to present the information in a more concise manner to avoid overwhelming the reader with excessive detail.

5.     Topic 2.1.2 and 2.1.3, the trapping with methanol and with 1,2-diol did not result in the isolation of the corresponding boronic acid. It is recommended to move these topics to the SI section and report only topic 2.1.4 in the main text. A few sentences should be sufficient to broadly explain the failure of isolation via methanol and 1,2-diol.

6.     Topic 2.1.5 and 2.1.6, purification protocol and deprotection step can be as considered as comment 5 because the synthetic details were already described in topic 5.4. The unsuccessful experiments should also be briefly mentioned in the main text and placed full detail the SI section to avoid redundancy.

7.     Page 9, line 302, recheck texts in parenthesis if questions marks are extraneous.

8.     The entire topic 2.2 on NMR analyses of intermediate 6 and target compound 7 is overwhelming for the galley. It would be more appropriate to move most of the data to the SI section, while crucial intermediates could be elaborated in the galley.

Alternatively to moving NMR material to the galley is splitting the paper into one on NMR; in other words, publish all of the NMR work in a journal such as Organic Magnetic Resonance or other structurally related journal.   Then publish the highlhights and biological studies in a separate journal.  While comprehensive accounts are preferred to multiple fragmented reports, when the topic is broad it may be better to divide the work into separate papers.  The work will have greater impact in that manner.  The present document may be off-putting to anyone interested in the biology and medicine.  Those interested in structural features may not find this work. 

Conclusions: This study reported the synthetic methodology and comprehensive characterization of the target. Preliminary biological data provide an initial insight into the properties of the borylated product for future research. The document reports a tour de force in terms of comprehensiveness, but at the same time, reads like a PhD thesis.  Rarely will the detailed NMR tables be of interest to readers, I feel; I realize this is not what the author would like to hear given how much work went into this body of work.   The findings of this research are well-suited for publication in Molecules, but the author should think clearly whether putting everything on all topics pertaining to these compounds in one manuscript serves her, the science, and the scientific community very well.  A separate spectroscopy-related paper could be recommended to present the spectroscopic studies. 

Author Response

I would like to thank the Reviewer for doing such an amazingly detailed review work. It’s massively appreciated. I have modified the manuscript according to their suggestions, including dividing the manuscript up into two manuscripts to allow an easier read and focus on Synthesis and Bio in one manuscript and on Spectroscopic Characterisation in another. This division allows for a more focused approach and the reader doesn’t get lost in all the information. Thank you for this

Please see below the answers to the points raised by the Reviewers.

The diethanolamine boronate ester derivative of monosaccharide D-galactose was diastereoselectively synthesized through the efficient one-pot two-step procedures, offering a quicker pathway to obtain the desired borylated product with a greener purification process. The resulting product was characterized in detail and biological studies were conducted to evaluate the inhibitory activity of the synthesized borylated product on glycosidase enzymes. The results showed that the product did not exhibit any significant inhibition of glycosidase enzymes. However, the borylated product demonstrated low cytotoxicity against tumor and normal cell lines and was identified as a promising candidate for cancer treatment using BNCT.

Comments.

  1. The introduction is rather broad, a more focused statement that highlights the importance of glycosidase modulators against diseases would allow readers to better understand the importance of the borylated compound. Additionally, an explanation of how switch on/switch off BNCT agents mediate therapy would enhance the readers' comprehension. This was done as suggested.
  2. The study claims that the synthetic method in this work achieves the highest yield for the installation of boronic acids on high Fsp3 index substrates. To support this claim, the authors should provide comparative data with conventional methods.
  3. Topic 2.1, I would suggest shortening this section by moving details about purification, storing methods, and solvent choices to the Supporting Information (SI) section. This will help streamline the main text and allow readers to focus on the key aspects of your research. This was done as suggested.
  4. Topic 2.1.1, trapping optimization, the author provided valuable background information. However, it would be beneficial to present the information in a more concise manner to avoid overwhelming the reader with excessive detail. This was done as suggested.
  5. Topic 2.1.2 and 2.1.3, the trapping with methanol and with 1,2-diol did not result in the isolation of the corresponding boronic acid. It is recommended to move these topics to the SI section and report only topic 2.1.4 in the main text. A few sentences should be sufficient to broadly explain the failure of isolation via methanol and 1,2-diol. This was done as suggested.
  6. Topic 2.1.5 and 2.1.6, purification protocol and deprotection step can be as considered as comment 5 because the synthetic details were already described in topic 5.4. The unsuccessful experiments should also be briefly mentioned in the main text and placed full detail the SI section to avoid redundancy. This was done as suggested.
  7. Page 9, line 302, recheck texts in parenthesis if questions marks are extraneous. AMended
  8. The entire topic 2.2 on NMR analyses of intermediate 6 and target compound 7 is overwhelming for the galley. It would be more appropriate to move most of the data to the SI section, while crucial intermediates could be elaborated in the galley. Amended

Alternatively to moving NMR material to the galley is splitting the paper into one on NMR; in other words, publish all of the NMR work in a journal such as Organic Magnetic Resonance or other structurally related journal.   Then publish the highlhights and biological studies in a separate journal.  While comprehensive accounts are preferred to multiple fragmented reports, when the topic is broad it may be better to divide the work into separate papers.  The work will have greater impact in that manner.  The present document may be off-putting to anyone interested in the biology and medicine.  Those interested in structural features may not find this work.  I have produced an NMR manuscript and simplified/clarified the current one.

Conclusions: This study reported the synthetic methodology and comprehensive characterization of the target. Preliminary biological data provide an initial insight into the properties of the borylated product for future research. The document reports a tour de force in terms of comprehensiveness, but at the same time, reads like a PhD thesis.  Rarely will the detailed NMR tables be of interest to readers, I feel; I realize this is not what the author would like to hear given how much work went into this body of work.   The findings of this research are well-suited for publication in Molecules, but the author should think clearly whether putting everything on all topics pertaining to these compounds in one manuscript serves her, the science, and the scientific community very well.  A separate spectroscopy-related paper could be recommended to present the spectroscopic studies.